# Limited induction of polyfunctional lung-resident memory T cells against SARS-CoV-2 by mRNA vaccination compared to infection

Daan K. J. Pieren [1], Sebastián G. Kuguel[1], Joel Rosado[2], Alba G. Robles[1], Joan Rey-Cano[1], Cristina Mancebo[1], Juliana Esperalba [3], Vicenç Falcó [1], María J. Buzón [1] & Meritxell Genescà [1] ✉

Resident memory T cells ($T_{RM}$) present at the respiratory tract may be essential to enhance early SARS-CoV-2 viral clearance, thus limiting viral infection and disease. While long-term antigen-specific $T_{RM}$ are detectable beyond 11 months in the lung of convalescent COVID-19 patients, it is unknown if mRNA vaccination encoding for the SARS-CoV-2 S-protein can induce this frontline protection. Here we show that the frequency of CD4$^+$ T cells secreting IFNγ in response to S-peptides is variable but overall similar in the lung of mRNA-vaccinated patients compared to convalescent-infected patients. However, in vaccinated patients, lung responses present less frequently a $T_{RM}$ phenotype compared to convalescent infected individuals and polyfunctional CD107a$^+$ IFNγ$^+$ $T_{RM}$ are virtually absent in vaccinated patients. These data indicate that mRNA vaccination induces specific T cell responses to SARS-CoV-2 in the lung parenchyma, although to a limited extend. It remains to be determined whether these vaccine-induced responses contribute to overall COVID-19 control.

The COVID-19 pandemic continues, and many countries face multiple resurgences. While vaccines to limit SARS-CoV-2 infection rapidly emerged, providing high protection from COVID-19, more insight into the mechanisms of protection induced by available vaccines is still needed. The level of vaccine-induced neutralizing antibodies has been shown to correlate with protection from symptomatic infection; however, predicted antibody-mediated vaccine efficacy declines over time[1]. Moreover, many viral variants of concern (VOC) can significantly evade humoral immunity, yet cellular responses induced by vaccines show strong cross-protection against these variants[2,3], supporting the idea that cellular responses largely contribute to disease control[4]. In fact, pre-existing cross-reactive memory T cells and early Nucleocapsid (N) responses against coronaviruses are associated with protection from SARS-CoV-2 infection[5,6].

Further, SARS-CoV-2 infection induces robust cellular immunity detectable beyond 10 months after infection in peripheral blood[7], and as $T_{RM}$ in the lung[8], the number of SARS-CoV-2-specific $T_{RM}$ in the lung correlates with clinical protection[9]. Vaccination against SARS-CoV-2 using BTN162b2 (Pfizer/BioNTech) and mRNA-1273 (Moderna) vaccines has been reported to induce CD4$^+$ and CD8$^+$ T cell responses in peripheral blood[10,11]. Moreover, the IFNγ T cell response to SARS-CoV-2 S-peptides, one of the main antiviral factors measured as a readout, further increased after boosting[11]. However, current studies only address vaccine-induced SARS-CoV-2-specific T cell responses in peripheral blood, and whether mRNA vaccines also elicit SARS-CoV-2-specific long-term $T_{RM}$ cells in the lung remains to be established.

To this end, we determined the presence of SARS-CoV-2-specific CD4$^+$ and CD8$^+$ T cells in 30 paired peripheral blood and lung cross-sectional samples from (I) uninfected unvaccinated individuals (Ctrl,

[1]Infectious Diseases Department, Vall d'Hebron Institut de Recerca (VHIR), Vall d'Hebron Hospital Universitari, Vall d'Hebron Barcelona Hospital Campus, Passeig Vall d'Hebron 119-129, 08035 Barcelona, Spain. [2]Thoracic Surgery and Lung Transplantation Department, Vall d'Hebron Institut de Recerca (VHIR), Vall d'Hebron Hospital Universitari, Vall d'Hebron Barcelona Hospital Campus, Passeig Vall d'Hebron 119-129, 08035 Barcelona, Spain. [3]Respiratory Viruses Unit, Microbiology Department, Vall d'Hebron Institut de Recerca (VHIR), Vall d'Hebron Hospital Universitari, Vall d'Hebron Barcelona Hospital Campus, Passeig Vall d'Hebron 119-129, 08035 Barcelona, Spain. ✉e-mail: meritxell.genesca@vhir.org

$n = 5$), (II) unvaccinated long-term SARS-CoV-2 convalescent individuals (Inf, $n = 9$, convalescent for a median of 304 days [183–320 IQR]), III.) uninfected and long-term two- or three-dose vaccinated individuals (LT, $n = 10$, a median of 208 days [198–261] after the second or third dose), and (IV) uninfected and short-term three- or four-dose vaccinated individuals (ST, $n = 6$, a median of 53 days [45–56] after the third or fourth dose). Whereas our data showed that S-specific T cells could be detected in the lung of mRNA-vaccinated individuals up to 10 months after immunization, lung responses in vaccinated patients presented less frequently a $T_{RM}$ phenotype and polyfunctional $T_{RM}$ expressing IFNγ and CD107a were essentially absent compared to convalescent patients.

## Results

### Cohort characteristics

Paired cross-sectional peripheral blood and healthy tissue areas obtained from patients undergoing lung resection for different reasons (mostly suspicion of cancer) were studied. A schematic summary of patients included in this study is shown in Fig. 1a, and additional patient characteristics are summarized in Supplementary Table 1. In order to confirm the SARS-CoV-2 status of each patient, we analyzed levels of total immunoglobulin (Ig) against N protein and IgG against Spike (S) protein, which discriminated Ctrl (negative for anti-N Ig and anti-S IgG), Inf patients (positive for anti-N Ig and anti-S IgG) and vaccinated groups (negative for anti-N Ig and positive for anti-S IgG; Fig. 1a and Supplementary Table 1). Furthermore, the viral neutralization titer was determined against the SARS-CoV-2 Omicron variant using a pseudovirus neutralization assay and, as expected[10,11], a positive correlation between neutralization and S-IgG titers was detected (Spearman $r = 0.69$, $P = 0.0005$; Supplementary Fig. 1a). In addition to the absence of neutralization capacity of the Omicron variant in the plasma of the Ctrl group, two out of seven patients (28%) in the Inf group and one out of ten patients (10%) in the LT group failed to neutralize the virus, whereas all patients in the ST group were able to neutralize this variant (Supplementary Table 1). Patients included in our study were, with two exceptions, mostly middle-aged (50–65 years, $n = 11$) and older (66–81 years, $n = 17$) adults. Whereas we found a negative correlation between older age and neutralizing capacity for the Inf group (Spearman $r = −0.88$, $P = 0.01$; Supplementary Fig. 1b) we did not detect this for the LT vaccinated group (Spearman r $= −0.53$, $P = 0.15$; Supplementary Fig. 1c). This relationship was less evident between age and S-IgG titers in the Inf group (Supplementary Fig. 1d), yet more pronounced in the LT group (Spearman r $= −0.79$, $P = 0.016$; Supplementary Fig. 1e), similar to findings in larger cohorts[10]. Furthermore, S-IgG titers from all groups combined negatively correlated with sample timing (Spearman $r = −0.58$, $P = 0.006$; Supplementary Fig. 1f), a correlation that was also observed for total Ig against N in the Inf group (Spearman $r = −0.88$, $P = 0.009$; Supplementary Fig. 1g), which agrees with the decay of antibody titers shown previously[10–12].

### Recent mRNA booster vaccination elicits S-specific CD4+ T cells similar to convalescent infection

To address cellular immune responses, we stimulated fresh peripheral blood mononuclear cells (PBMC) and lung-derived cellular suspensions with overlapping Membrane (M), N, and S peptide pools and determined the intracellular expression of IFNγ, interleukin (IL)-4, and IL-10, along with the degranulation marker CD107a in CD4+ and CD8+ T cells (gating strategy in Supplementary Fig. 2), as previously described[8]. We found detectable circulating IFNγ-secreting Ag-specific CD4+ T cells responding to all proteins in the blood of Inf patients, which was significantly higher compared to the Ctrl, LT, and ST groups for M and N peptides (Fig. 1b, c). However, both the Inf and ST groups showed higher frequencies of S-specific IFNγ+ CD4+ T cells compared to the Ctrl group (Fig. 1c). Moreover, in contrast to

recently boosted ST patients, only three out of ten LT patients showed detectable frequencies of S-specific CD4+ T cells in blood (Fig. 1c). The frequencies of IFNγ+ CD8+ T cells detected were minimal for each of the groups against any of the proteins, including for the LT and ST groups against S peptides (Fig. 1c). Expression of IL-4, IL-10, and CD107a by T cells showed, in general, high variability, limiting the detection of differences (Supplementary Fig. 3). Nonetheless, S-specific degranulating CD107a+ CD8+ T cells were overall more frequent in the LT group compared to the Ctrl group ($P = 0.022$; Supplementary Fig. 3). Together, these data indicate that M, N, and S-peptide specific IFNγ+ CD4+ T cell responses can be readily detected in blood months after resolving natural SARS-CoV-2 infection and that these responses require a recent mRNA vaccine booster-dose against SARS-CoV-2 to elicit similar frequencies against the S protein in most vaccinated individuals.

### mRNA vaccination induces S-specific CD4+ T cells in the lung

As reported previously in ref. 8, we here found that robust IFNγ+ CD4+ T cells can be detected in the lung against M, N, and S peptides up to 12 months after a mild or severe natural infection with SARS-CoV-2 (Fig. 2a, b). Interestingly, whereas M and N-specific IFNγ+ CD4+ T cell frequencies were significantly higher in the Inf group compared to the Ctrl, LT, and ST groups, these differences were not observed for S-specific responses (Fig. 2a, b). LT and ST groups showed presence of S-specific IFNγ+ CD4+ T cells in the lung in half of the LT patients (5/10 patients) and most ST patients (5/6 patients) and their frequencies were comparable to levels detected in Inf patients, although statistical significance was not reached compared to the Ctrl group (Fig. 2b). In contrast to CD4+ T cells, the level of CD8+ T cells producing IFNγ after stimulation with M, N, or S peptides was variable within each group and did not result in significant differences between the groups, indicating that natural infection nor vaccination elicit a robust IFNγ positive CD8+ T cell response in human lung (Fig. 2b). Furthermore, induction of lung anti-SARS-CoV-2-specific T cell responses involving expression of IL-4, IL-10, and CD107a did not differ between groups (Supplementary Fig. 4a). Of note, we detected negative correlations between patient age within the Inf group and the frequency of S-specific degranulating CD4+ and CD8+ T cells in the lung (Spearman $r = −0.76$, $P = 0.025$ and Spearman $r = −0.77$, $P = 0.021$ respectively, Supplementary Fig. 4b).

When we compared the magnitude of S-specific T cells in paired blood and lung samples, we found increased frequencies of IFNγ+ CD4+ T cells in the lungs of patients from the Inf group compared to blood ($P = 0.039$, Fig. 2c). The same trend was observed for IFNγ+ CD4+ T cells of LT and ST patients when both groups were pooled ($P = 0.06$, Supplementary Fig. 5a), although this increase was more variable as only 10 out of 16 LT and ST-vaccinated patients showed an increase, in contrast to eight out of nine Inf patients. The CD8+ T cell subset did not show clear differences regarding the IFNγ+response between blood and lung (Fig. 2c and Supplementary Fig. 5a), nor any of the T cell subsets for any other function, which were highly variable (Supplementary Fig. 5b, c).

Of note, stimulation with SARS-CoV-2 peptides consisted of 15-mer peptide pools, which may be less optimal than shorter 9/10-mer peptides for HLA class I binding[3], thereby possibly underestimating CD8+ T cell responses. Thus, we additionally compared the CD4+ and CD8+ T cell IFNγ response in PBMCs ($n = 8$) and lung samples ($n = 5$) after stimulation with 15-mer and 9/10-mer S-peptide pools. The IFNγ response of CD8+ T cells did not differ between 15-mer or 9/10-mer peptide pools in blood or lung, although there was some variability (Supplementary Fig. 6). Moreover, in blood but not lung, the CD4+ T cell response was significantly lower when stimulated with 9/10-mer peptides compared to 15-mer peptides. Thus, with the exception of some individuals, stimulation with 15-mer peptides detected the same or higher T cell response compared to shorter

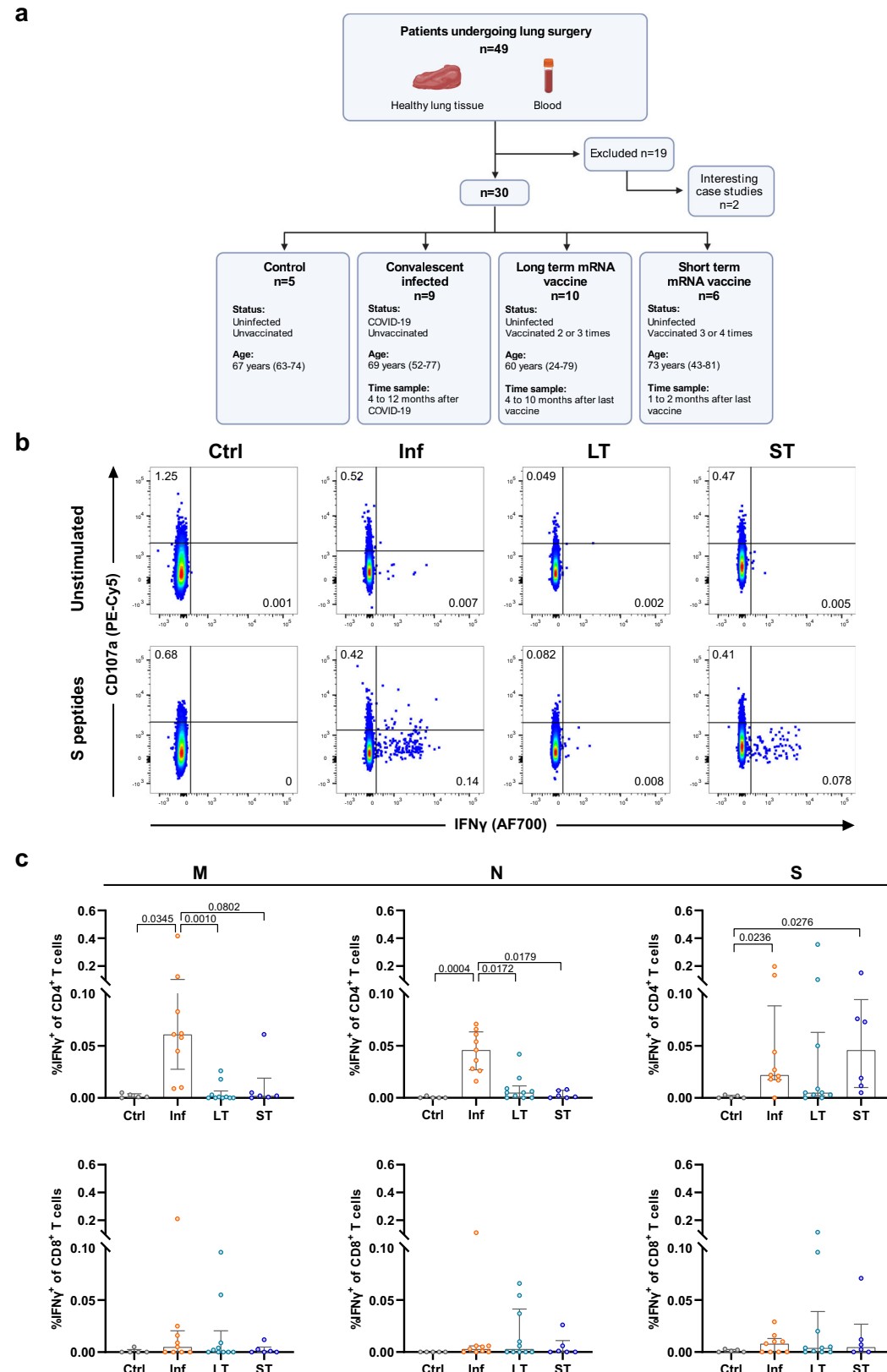

peptides, corroborating the overall results detected in our patients. Together, our data indicate that S-specific CD4+ T cell responses are detectable in the lung of uninfected-vaccinated patients, suggesting that mRNA vaccination against SARS-CoV-2 may potentially elicit tissue-localized protective T cell responses already after the second mRNA vaccine dose.

## Limited induction of tissue-resident memory T cells by mRNA vaccination

The presence of T$_{RM}$ cells may provide a better correlate of protection from disease in SARS-CoV-2-infected individuals and these cells are characterized by expression of CD69 and/or CD103[8,9,13]. Moreover, T$_{RM}$ cells require downregulation of the transcription

**Fig. 1 | SARS-CoV-2-specific T cell responses in peripheral blood from convalescent and vaccinated patients. a** Schematic overview of patients included in this study and characteristics of the study groups. **b** Representative flow-cytometry plots showing CD4+ T cells expressing CD107a and IFNγ after exposure of whole PBMCs to S-peptide pools or left unstimulated for each of the four groups included in this study (complete gating strategy is shown in Supplementary Fig. 2a). **c** Comparison of the net frequency (background subtracted) of IFNγ+ cells within

CD4+ (upper) and CD8+ (lower) T cell subsets after stimulation of PBMCs with any of the three viral peptide pools (membrane (M), nucleocapsid (N), and spike (S) peptides). Data were shown as median ± IQR, where each dot represents an individual patient for each group (Ctrl, control, $n = 5$; Inf convalescent infected, $n = 9$; LT vaccine 2/3 doses, $n = 10$, and ST, vaccine 3/4 doses, $n = 6$). Statistical significance was determined by Kruskal–Wallis test (with Dunn's post-test, two-sided). Source data are provided as a Source Data file.

factor T-bet for expression of CD103 and their formation and survival at tissue sites[14]. In order to assess if S-specific T cell responses detected in the lung of vaccinated patients indeed expressed a $T_{RM}$ phenotype, we analyzed the expression of CD69 and CD103 by lung SARS-CoV-2-specific CD4+ and CD8+ T cells, which we classified as CD69- (non-$T_{RM}$), CD69+ ($T_{RM}$) and a subset within CD69+ cells expressing CD103+ ($T_{RM}$ CD103+) (Fig. 3 and Supplementary Fig. 2b for gating strategy). Whereas lung biopsies were thoroughly perfused ex vivo to clear remaining blood, we additionally assessed the expression of T-bet to assure that CD69 in the lung was associated with a $T_{RM}$ phenotype and not to activation of CD69- T cells or the product of residual blood in the lung. Both CD69+ $T_{RM}$ and CD103+ $T_{RM}$ subsets did not show T-bet expression across all patient groups, whereas a fraction of CD69- non-$T_{RM}$ cells presented T-bet expression (Supplementary Fig. 2b), suggesting the association to tissue residency when absent[8]. S-specific CD4+ T cells from the Inf group showed higher frequencies of IFNγ+ cells within the CD69+ and CD103+ $T_{RM}$ phenotypes (Fig. 3a, b), with statistical significance, reached for the overall CD69+ $T_{RM}$ fraction compared to the non-$T_{RM}$ fraction. No significant differences were detected for CD103+ $T_{RM}$ cells against S-peptides in any of the groups. Furthermore, in the Inf group, a trend was observed for CD4+ $T_{RM}$ responses to M peptides and statistical significance was reached for CD8+ CD103+ $T_{RM}$ cells against N peptides compared to the non-$T_{RM}$ fraction (Supplementary Fig. 7a, b). Of note, a negative correlation was observed between IFNγ-secreting S-specific CD8+ CD103+ $T_{RM}$ cells and sample timing (Spearman $r = -0.82$, $P = 0.019$ Supplementary Fig. 7c). In addition, some patients in the LT and ST groups showed the modest presence of S-specific CD69+ $T_{RM}$ in their lungs (Fig. 3a, b). However, this response was highly heterogeneous and not statistically significant. These findings indicate that mRNA vaccination against SARS-CoV-2 is capable of inducing S-specific $T_{RM}$ in some, but not all, individuals and may also last long-term after the second vaccination.

To further confirm the resident nature of these Ag-specific T cells aside from phenotypical cellular markers, we performed chemoattraction assays with CCL19, CCL21, and S1P[15] to attract SARS-CoV-2 M, N, and S-specific T cells derived from either infected and vaccinated patients ($n = 2$) or non-infected and LT vaccinated patients ($n = 2$) out of lung tissue blocks. As shown in the representative flow-cytometry plots, the majority of SARS-CoV-2-specific T cells remained within the tissue and did not migrate towards the attracting signals (Supplementary Fig. 8a). In most patient lung samples, chemoattractants increased the number of emigrant CD3+ T cells, and we hardly detected any specific T cells in the emigrant fraction (Supplementary Fig. 8b, blue lines). As a control, PBMCs ($n = 2$) were placed in the same system, resulting in CD3+ T cell emigration and the detection of SARS-CoV-2-specific CD3+ T cells in both the emigrated and non-emigrated fractions (Supplementary Fig. 8a, b, red lines). Aside from these T cell-migration assays, we also assessed whether circulating SARS-CoV-2-specific T cells of vaccinated patients displayed a homing phenotype compatible with homing to the lung[8]. The absence of CCR7 expression indicates effector memory T cells migrating to tissues, while expression of CXCR3 may indicate T cells with a potential to infiltrate in the lung parenchyma, among other tissues[8,16,17]. In the blood of vaccinated individuals with an S-peptide response, including LT

($n = 2$), ST ($n = 5$), and three additional samples of convalescent infected and recently vaccinated (ST, <1 week ago) individuals, we found that the majority of IFNγ+ CD4+ T cells (>80%) lacked expression of CCR7, indicating tissue migration, from which a significant fraction expressed CXCR3 (Supplementary Fig. 8b, c). Together, these data indicate that SARS-CoV-2-specific T cells expressing $T_{RM}$-associated markers can be established as long-term $T_{RM}$ cells in the lung of both, convalescent infected and mRNA-vaccinated patients.

## Overall functional T cell response of lung and blood compartments

To better gain insight into the overall S-specific response by each group, including all functions and considering lung-$T_{RM}$ phenotypes, we represented S-specific CD4+ and CD8+ T cell subsets as donut charts displaying the mean frequency of responses including all individuals (responders and non-responders, Fig. 4). This way, a dominance of IFNγ-secreting CD4+ T cells was particularly associated to the two $T_{RM}$ phenotypes in the Inf and, to a lesser extent, in the LT and ST patients (Fig. 4a). Further, S-specific responses within non-$T_{RM}$ and blood CD4+ T cells were functionally similar and in general dominated by IFNγ and IL-4 secretion (Fig. 4a). In contrast, degranulation characterized the majority of lung S-specific CD8+ T cells from Inf individuals (Fig. 4b), which correlated negatively with older age for the $T_{RM}$ fractions (Spearman $r = -0.88$, $P = 0.006$ for both CD103 positive and negative, Supplementary Fig. 7d). Degranulation was a major function in blood from the LT group, while IL-4 was predominant in the Inf and the ST groups (Fig. 4b). Last, in general, CD8+ T cell responses considering all functions were of higher magnitude in LT patients, reaching statistical significance for blood responses in comparison to the Ctrl group, as shown in the adjoin bar graph on the right (Fig. 4b).

## Polyfunctional S-specific IFNγ+CD107a+ CD4+ T cells are absent in vaccinated patients

We previously detected a low but consistent polyfunctional IFNγ+CD107a+ T cell response mostly associated with the $T_{RM}$ fraction in convalescent-infected patients[8]. We, therefore, investigated whether mRNA vaccination against SARS-CoV-2 would also induce such S-specific polyfunctional responses in blood or lung (Fig. 5a, b). Indeed, increased frequencies of polyfunctional IFNγ+CD107a+ CD4+ T cells were detected in blood from the Inf group against N peptides compared to the Ctrl group, but not against M- and S-peptides. Interestingly, a trend towards higher frequencies of S-specific polyfunctional CD4+ T cells was observed for the ST group (Fig. 5a). Likewise, circulating polyfunctional S-specific CD8+ T cells were enhanced in LT individuals compared to the Ctrl group (Fig. 5a). In fact, if the LT and ST groups were pooled, then both CD4+ and CD8+ T cells reached significance compared to Ctrl samples ($P = 0.033$ for CD4+ and $P = 0.023$ for CD8+). In the lung, the frequency of polyfunctional IFNγ+CD107a+ cells present in total CD4+ and CD8+ T cells were generally increased in the Inf group against M and N peptides compared to the Ctrl, LT, and ST groups, reaching (trends towards) statistical significance (Fig. 5b). While a high degree of variability was observed among vaccinated patients, polyfunctional S-specific T cells were detected in some individuals (Fig. 5b). Remarkably, S-specific CD4+ polyfunctional CD103+ $T_{RM}$ cells were virtually absent in the LT and ST groups, while being present in the majority

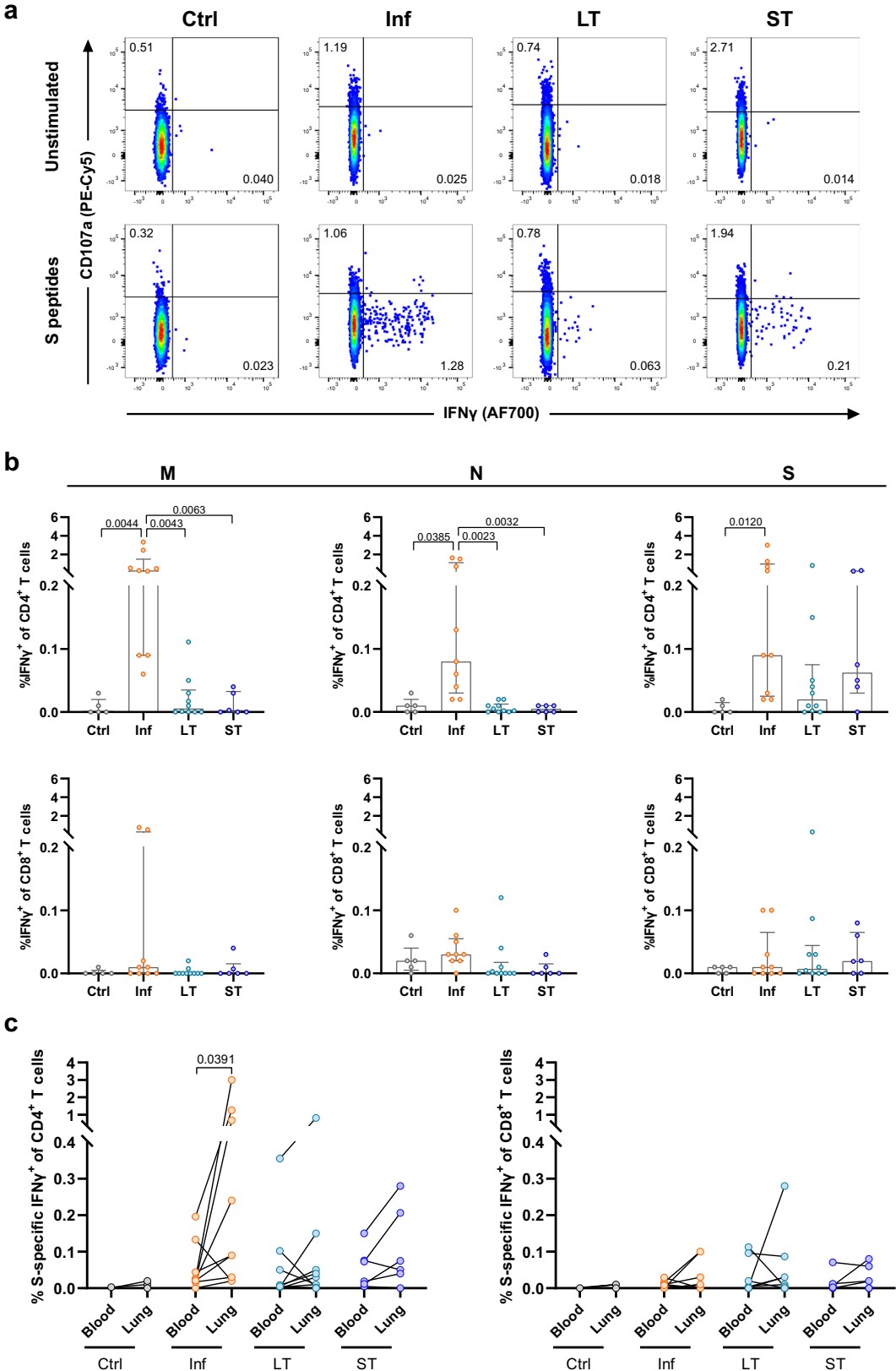

of the lungs of patients from the Inf group (Fig. 6). Furthermore, the frequency of S-specific polyfunctional CD4⁺ T cells in the CD69⁺ T$_{RM}$ cells was higher in the Inf group compared to the Ctrl and LT groups (Fig. 6). Together, these data indicate that both short- and long-term vaccination do not induce S-specific IFNγ⁺CD107a⁺ CD103⁺ T$_{RM}$ cells in the lung.

## Anti-SARS-CoV-2 T cell response dynamics in two cases of interest

Last, considering the uniqueness of analyzing immune responses in paired blood and lung parenchyma samples and recent studies detailing changes in T cell responses in infected individuals already vaccinated and vice versa[18], we highlight two patients that were

**Fig. 2 | SARS-CoV-2-specific lung T cell responses from convalescent and vaccinated patients and comparison between tissue compartments.**
**a** Representative flow-cytometry plots showing CD4$^+$ T cells expressing CD107a and IFNγ after exposure of single-cell suspensions of lung tissue to S-peptide pools or left unstimulated for each of the four groups included in this study (complete gating strategy is shown in Supplementary Fig. 2b). **b** Comparison of the net frequency (background subtracted) of IFNγ$^+$ cells within CD4$^+$ (upper) and CD8$^+$ (lower) T cell subsets after exposure of lung single-cell suspensions to any of the three viral peptide pools (membrane (M), nucleocapsid (N), and spike (S) peptides).

**c** Comparison of the net frequency of IFNγ$^+$ cells within CD4$^+$ (left) and CD8$^+$ (right) T cell subsets in paired blood and lung samples of each group after exposure to S-peptide pools. Data in bar graphs are shown as median ± IQR, where each dot represents an individual patient for each group (Ctrl, control, $n = 5$; Inf, convalescent infected, $n = 9$; LT, vaccine 2/3 doses, $n = 10$, and ST, vaccine 3/4 doses, $n = 6$). Statistical significance was determined by **b** Kruskal–Wallis test (with Dunn´s post-test, two-sided) or **c** Wilcoxon test (two-sided). Source data are provided as a Source Data file.

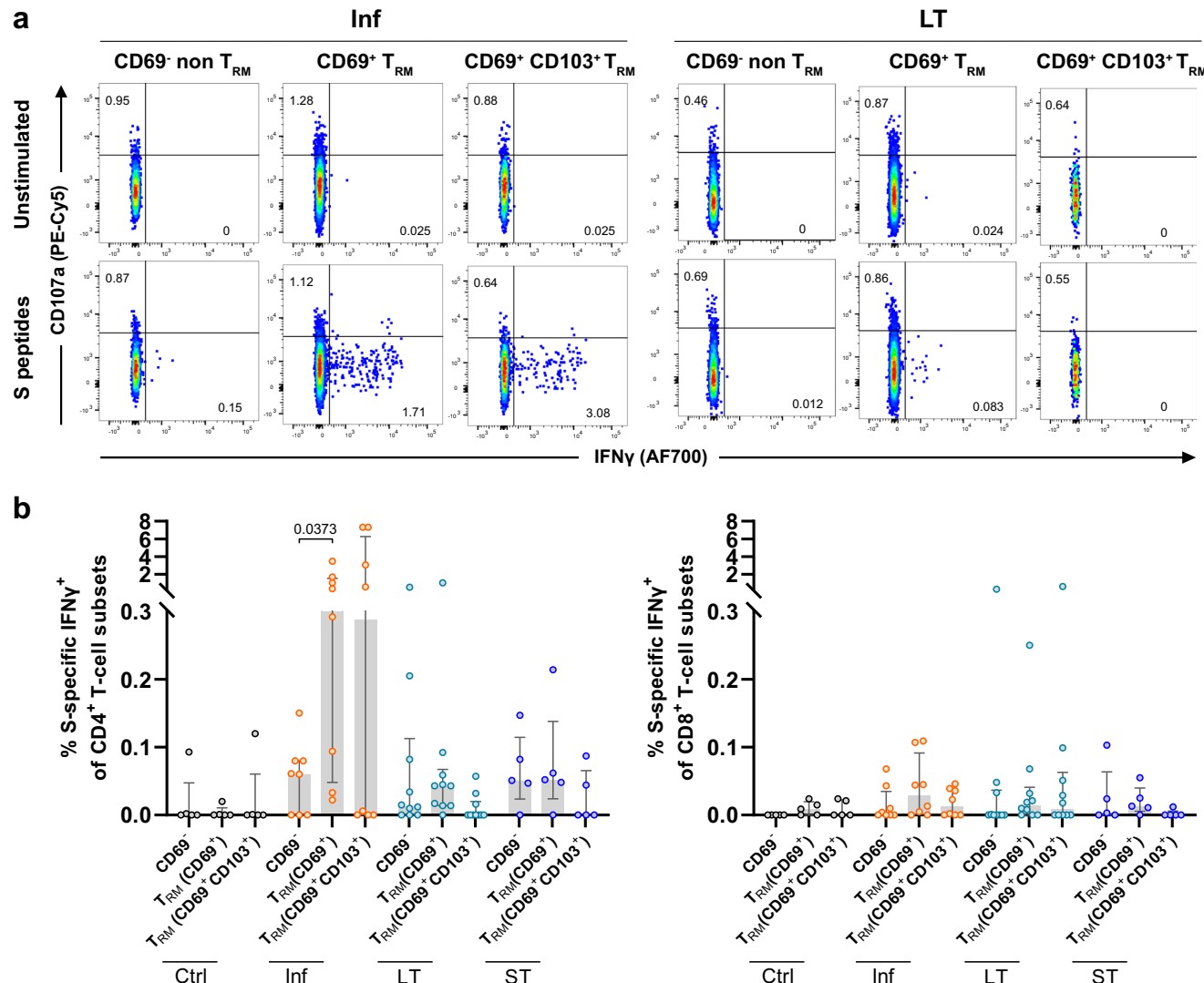

**Fig. 3 | Frequency of Spike-specific T$_{RM}$ cells in the lung. a** Representative flow-cytometry plots showing three subsets of CD4$^+$ T cells present in the lung: CD69$^-$ non-T$_{RM}$, CD69$^+$ T$_{RM}$, and CD69$^+$CD103$^+$ T$_{RM}$ cells expressing CD107a and IFNγ after exposure of single-cell suspensions of lung tissue to S-peptide pools or left unstimulated for an Inf and an LT patient. **b** Comparison of the net frequency of S-specific IFNγ$^+$ cells within the three (non-) T$_{RM}$ cell subsets present in the lung for

each group. Data in bar graphs are shown as median ± IQR, where each dot represents an individual patient for each group (Ctrl, control, $n = 5$; Inf, convalescent infected, $n = 8$; LT, vaccine 2/3 doses, $n = 10$, and ST, vaccine 3/4 doses, $n = 5$). Statistical significance was determined by the Friedmann test (with Dunn´s post-test, two-sided) for the difference between the cellular subsets within each patient group. Source data are provided as a Source Data file.

discarded due to not fitting inclusion criteria, yet these patients bring interesting data to the study.

HL174 was a patient in their fifties who received the third mRNA-1273 vaccine boost and, five days after, tested positive by PCR. We analyzed paired tissue samples 30 days after the boost/infection event (Supplementary Fig. 9a). This patient had a neutralization titer of 1740 IU/mL against omicron, and had detectable IgG and Ig titers against S and N proteins (>800 AU/mL and 1.23 index, respectively). When comparing T cell responses from blood and lung tissue, a much

higher IFNγ-response was observed in the lung, in particular against the N protein, which already contained responding cells with a T$_{RM}$ phenotype (Supplementary Fig. 9b, c). In contrast, in blood, degranulation was enhanced mostly against S but also M protein and some proportion of IL-10 secretion was detected against all proteins (Supplementary Fig. 9b).

Patient HL162, who was in their early seventies, first presented with mild COVID-19 disease and received three doses of the mRNA-1273 vaccine several months after. From this patient, we obtained

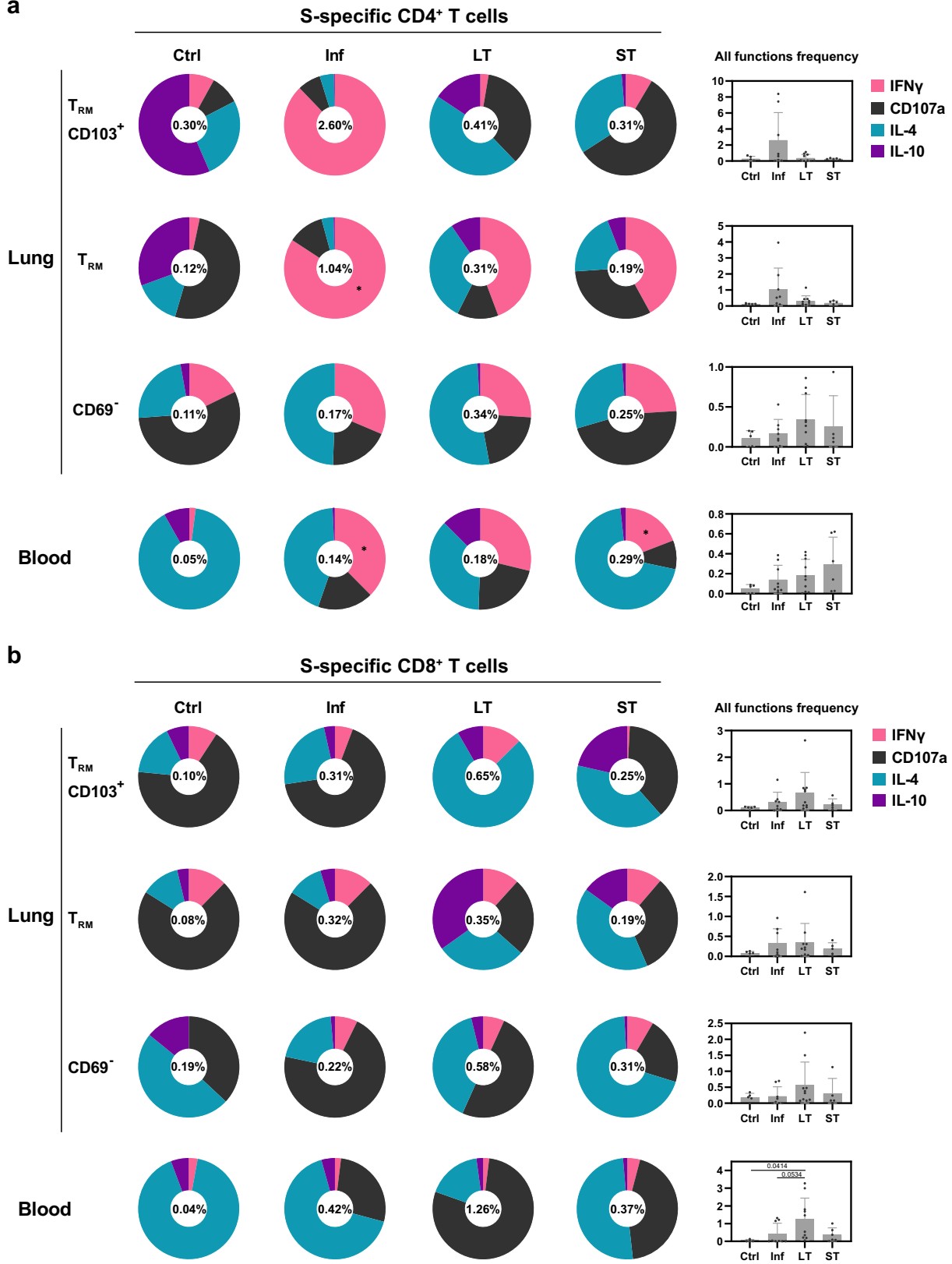

paired samples 3.7 months after infection, as well as 1.3 months after the third vaccine dose due to a second intervention for lung carcinoma, corresponding to a year after the initial infection (Supplementary Fig. 10a). Neither of these two time points showed neutralization titers against omicron and antibody titers, instead of increasing after triple vaccination, decreased from 156 to 0 index for Ig against the N

protein and from 306.54 to 13.85 AU/mL for IgG against the S protein. The comparison of the tissue compartments after infection and after triple vaccination evidenced a concomitant strong decrease in T cell responses in blood and tissue (Supplementary Figs. 10b, c, 11a, b). However, IFNγ-secreting SARS-CoV-2 T cells against M and N proteins in the lung were better preserved from the original infection one year

**Fig. 4 | Overall functional T cell response of lung and blood compartments.**
**a, b** Donut charts displaying the net contribution of each functional marker (IFNγ, CD107a, IL-4, and IL-10) to the overall S-specific CD4⁺ (**a**) and CD8⁺ (**b**) T cell response within the lung resident and non-resident T cell subsets and in peripheral blood for each of the individual patient groups. Data represent the mean value of the net frequency of each function within the patient group, including both responders and non-responders. The frequency shown inside each donut chart represents the accumulated mean response of all functions for each group (Ctrl, control, $n = 5$; Inf, convalescent infected, $n = 8$; LT, vaccine 2/3 doses, $n = 10$, and ST, vaccine 3/4 doses, $n = 5$). Bar charts on the right show the mean of the total frequency considering all functions per group (mean ± SD). Statistical significance was determined by Kruskal–Wallis test (with Dunn´s post-test, two-sided) for the difference between each group. *$P < 0.05$. Source data are provided as a Source Data file.

later than were responses against the S protein enhanced due to vaccination (Supplementary Figs. 10b, c, 11a, b). Thus, while the lower respiratory tract compartment more faithfully represented $T_{RM}$ responses established already during the infection event one year earlier, responses in blood mostly vanished.

## Discussion

Comprehensive studies comparing the magnitude and duration of the T cell responses indicate similar magnitude after dual vaccination and after natural SARS-CoV-2 infection[4,11,18]. However, these results may not hold if we consider that the magnitude, the functional profile, and even the duration of these responses in the blood may not faithfully reflect responses in the respiratory tract[6,8,9,19]. In fact, the individual comparison between these two compartments among the S-peptide responding T cells from the different groups showed higher magnitude in the lung than in the blood, but also a different profile. A key difference, and the main driver of our study, was the establishment of long-term protection potentially mediated by $T_{RM}$ after vaccination, since the longevity of SARS-CoV-2 T cell responses remains a critical question[6]. In principle, $T_{RM}$ are established by mucosal infection since Ag, together with local signals, promote the recruitment and establishment of this memory response. In this sense, intramuscular vaccination with an adenovector vaccine in mice did not induce SARS-CoV-2-specific $T_{RM}$ in their lungs[20]. Thus, to induce potent resident immunity, vaccine strategies may need to either use live-attenuated Ag or employ mucosal routes. Consequently, the absence of vaccine-induced S-specific $T_{RM}$ could be expected in infection-naïve individuals. Still, recent data shows that a secretory IgA response was induced in ~30% of participants after two doses of a SARS-CoV-2 mRNA vaccine which, in addition, may play an important role in protection against infection[19]. While we detected S-specific IFNγ⁺ CD4⁺ T cell responses in the lung of vaccinated individuals, the proportion of these cells in the $T_{RM}$ phenotype was modest, in particular when considering CD103 expression. Further, the presence of polyfunctional IFNγ⁺CD107a⁺CD4⁺CD103⁺ $T_{RM}$ appeared to be restricted to the lungs of convalescent-infected patients only, while absent in the lungs of LT and ST-vaccinated patients (Fig. 7 for a summary of the main findings of this study).

Another difference in the comparison of the cellular immunity between SARS-CoV-2-infected convalescent and uninfected-vaccinated individuals is the broader and potentially stronger response induced by symptomatic infection. This is partially manifested by the fact that the overall magnitude of responses against M and N peptides are frequently higher than S peptides[8,21–23]. Of note, disease severity may impact both the magnitude and function of the T cell response against the different proteins[8,24,25]. On the other hand, we have observed that different proteins induce different functional profiles during acute infection, which may influence disease control[8]. S-specific immune responses may better support B cell and antibody generation via follicular helper T cells, which are instrumental to limiting infection[4,8]. Instead, responses against the N protein seem to more consistently induce polyfunctional antiviral T cells and these responses may be more conserved among other coronaviruses[5,8,26,27]. Indeed, preexisting SARS-CoV-2-specific T cell responses have been found in the blood of unexposed individuals[22,28]. In this sense, in our study, we detected a low level of preexisting T cell responses in our

control group mostly located in the lung, which may be in agreement with a recent report in tonsillar tissue[29]. Thus, another conclusion would be highlighting the interest in including other proteins beyond the spike, such as N sequences, which has been suggested before[5,8,23,27,30]. Last, in terms of duration, our study lacks longitudinal data to assess the dynamics in the different compartments, yet it is assumed that $T_{RM}$ phenotypes will contribute to long-term persistence[6,8,31]. In fact, the only patient for which we had longitudinal sampling after infection and after the third vaccine boost (Supplementary Fig. 10) demonstrated that even if vaccination fails to induce a systemic antibody response, a low-frequency SARS-CoV-2 T cell response directed to proteins from the original infection remains exclusively detectable in the lung as $T_{RM}$ a year later.

The overall CD8⁺ T cell response was enhanced in some but not all vaccinated patients and was in general low and dominated by degranulation. In fact, lung S-specific CD8⁺ $T_{RM}$ presented similar overall frequencies in vaccinated individuals when considering all functions. Further, the comparison between S-specific T cell responses to 9/10-mer or the longer 15-mer peptides indicated that, in general, T cell responses against SARS-CoV-2 spike protein in our study were mainly driven by CD4⁺ T cells, as reported in the literature[4]. Nonetheless, variability may occur as some individuals may encompass higher CD8⁺ T cell responses to shorter S peptides, which may warrant the use of shorter S-peptide pools or other methods, such as the use of activation-induced markers and tetramers loaded with SARS-CoV-2 immunodominant peptides[29,32] to give additional insight into the CD8⁺ T cell response against SARS-CoV-2. Considering the putative protective role of CD8⁺ T cells observed in animal models[31], further exploration of the CD8⁺ T cell response after mRNA vaccination is desirable.

In mice, intramuscular immunization with an mRNA vaccine against influenza has recently been shown to elicit CD69⁺ and CD69⁺CD103⁺ $T_{RM}$ cells in the lung, which could be further boosted by intra-nasal immunization[13]. We here show that aside from the phenotypical presence of CD69⁺ and CD69⁺CD103⁺ $T_{RM}$ cells in the lung of infected and vaccinated patients, these SARS-CoV-2-specific T cells remain in lung tissue even when exposed to chemoattractants to induce cell migration. Moreover, the absence of CCR7 expression and substantial expression of CXCR3 by S-specific CD4⁺ T cells in peripheral blood further indicate that these cells may migrate towards tissues such as the lung[8,16,17]. Overall, our data indicate that the majority of specific SARS-CoV-2 T cell responses detected in the lung parenchyma of infected and vaccinated patients are bona fide $T_{RM}$ cells.

We acknowledge that our study has several limitations. The small sample size for the different groups warrants further investigation with ideally larger cohorts. Finding patients that fulfilled our inclusion criteria was, however, challenging, and currently, patients meeting the criteria are very scarce. In addition, the majority of patients included in this study were middle or older aged adults with the oncologic disease. Whereas cancer treatment was not initiated in all but one of our patients and thus did not affect our findings, it has recently been shown that after SARS-CoV-2 vaccination, neutralizing antibody responses and T cell responses in the blood of patients with thoracic malignancies do not differ from individuals without cancer[33,34]. However, patients with thoracic cancer are at higher risk of developing severe COVID-19 disease[35,36] and, in general, T cell responses against

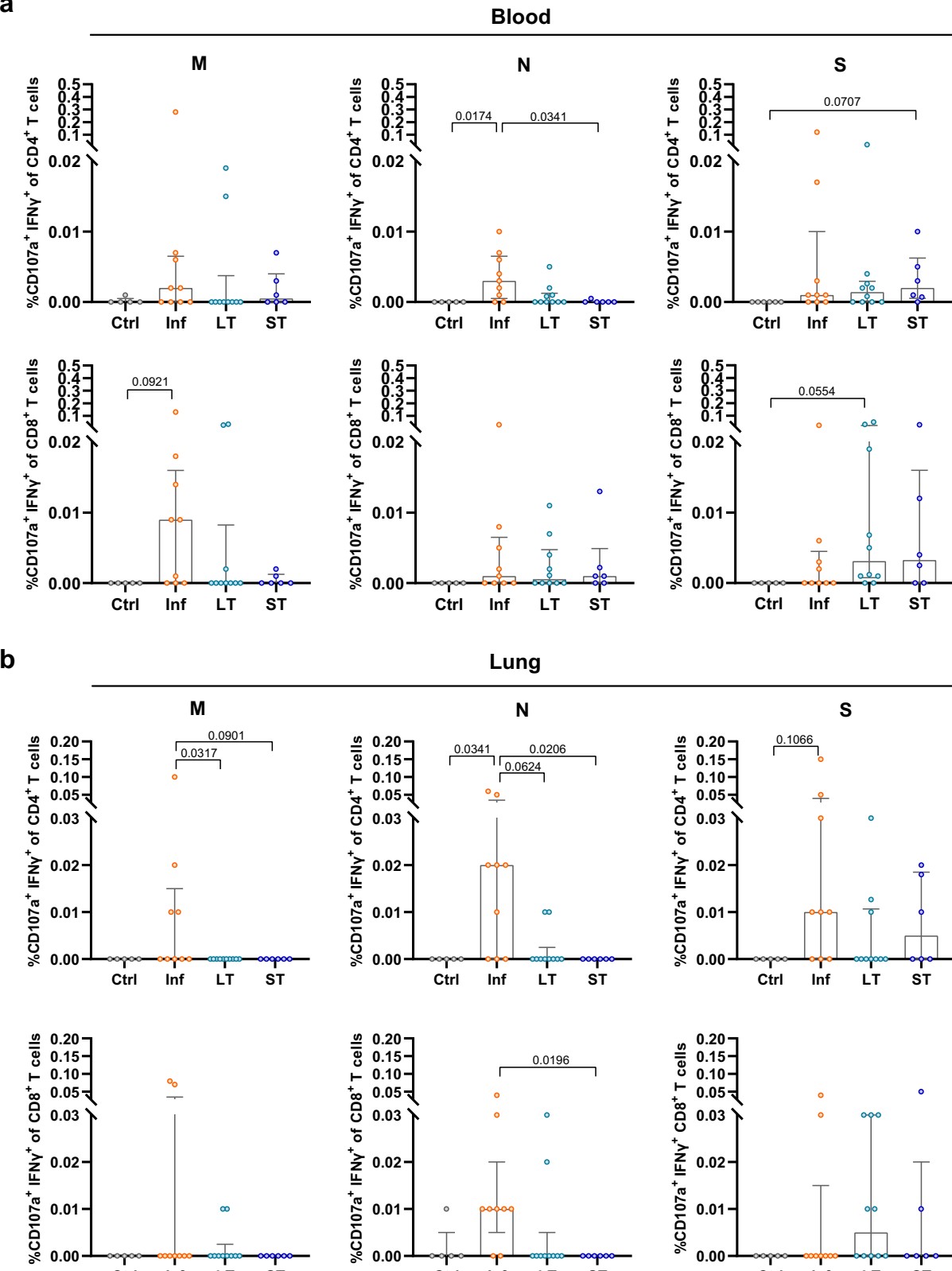

**Fig. 5 | Polyfunctional CD4⁺ and CD8⁺ T cell responses in blood and lung of convalescent and vaccinated patients. a**, **b** Comparison of the net frequency of polyfunctional CD107a⁺ IFNγ⁺ cells within CD4⁺ and CD8⁺ T cell subsets for each of the four groups after exposure of PBMCs (**a**) or single-cell suspensions of lung tissue (**b**) to any of the three viral peptide pools (membrane (M), nucleocapsid (N), and spike (S) peptides). Data in bar graphs are shown as median ± IQR, where each dot represents an individual patient for each group (Ctrl, control, $n = 5$; Inf, convalescent infected, $n = 9$; LT, vaccine 2/3 doses, $n = 10$, and ST, vaccine 3/4 doses, $n = 6$). Statistical significance was determined by was determined using Kruskal–Wallis test (with Dunn´s post-test, two-sided). Source data are provided as a Source Data file.

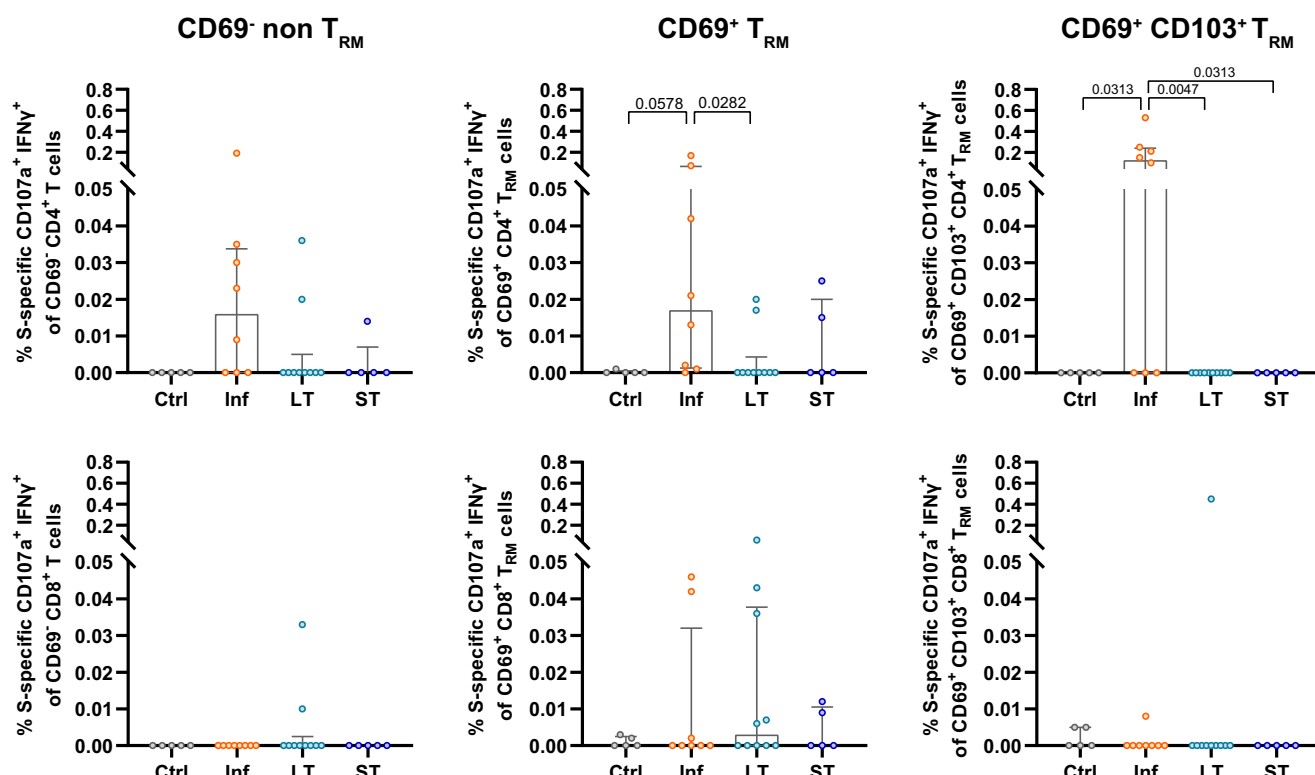

**Fig. 6 | Frequency of polyfunctional T cell responses against spike with a tissue-resident phenotype in the lung.** Comparison of the net frequency of S-specific polyfunctional CD107a$^+$ IFNγ$^+$ cells within lung CD4$^+$ (upper) and CD8$^+$ (lower) (non-) tissue-resident cell subsets (CD69$^-$ non-T$_{RM}$, CD69$^+$ T$_{RM}$, and CD69$^+$CD103$^+$ T$_{RM}$ cells) for each of the four patient groups after exposure of single-cell lung suspensions to S-peptide pools. Data in bar graphs are shown as median ± IQR, where each dot represents an individual patient for each group (Ctrl, control, $n = 5$; Inf, convalescent infected, $n = 8$; LT, vaccine 2/3 doses, $n = 10$, and ST, vaccine 3/4 doses, $n = 5$). Statistical significance was determined by Kruskal–Wallis test (with Dunn´s post-test, two-sided) for the difference between the groups. Source data are provided as a Source Data file.

SARS-CoV-2 from cancer patients after vaccination may be impaired[37], which may overall underestimate or add to variability in the immune responses across all groups of our study. Nonetheless, we still detected vaccine-induced S-specific CD4$^+$ T cells in our patients, which indicates that mRNA vaccination may even contribute to protection against COVID-19 in these potentially vulnerable patients. In addition, patients included in our study were mostly middle or older aged adults. In general, older age may influence the magnitude, duration, and variability of immune responses in response to vaccination[38,39] and even the establishment of T$_{RM}$[40] in distinct tissue compartments. Indeed, age was a factor that correlated with declined degranulation in the lung of our Inf patients, including within the T$_{RM}$ fraction. However, recent studies have also shown that boosting with SARS-CoV-2 mRNA vaccines elicits competent immune responses to SARS-CoV-2 and variants of concern in older adults[41,42], indicating that the responses we found in the lungs of vaccinated patients may possibly provide long-term protection. Further, the Inf group consisted of patients who recovered from mild or severe disease. Whereas age and underlying conditions were similar to the other groups, disease severity may have skewed frequencies of SARS-CoV-2-specific T cells towards the higher end. Still, considering their age and condition, any of these patients encountering a new SARS-CoV-2 infection would most likely develop a more serious COVID-19 event compared to the general population. In addition, the recently boosted ST group was sampled short term compared to the LT group, but enhancement of T cell responses would be better detected 5–10 days after boosting[10,11], which was a less likely time for scheduling surgery. Last, we did not assess the contribution of T cells targeting mutation regions to the total spike since we aimed to compare the strength and function of vaccinated and naturally

infected patients (these last groups were obtained during the first wave). However, the overall contribution of T cell responses to mutational regions/total spike responses has been reported to be low[18,43].

Overall, our results contribute to the understanding of disease protection mediated by current mRNA vaccines. While our data indicate, a more robust and broader cellular response in convalescent patients, S-specific T cells can be detected in the lung of vaccinated individuals to similar overall levels up to 10 months after immunization, highlighting the durability of this immune arm. Further, we detected increased levels of IFNγ$^+$ T cell responses in blood after a recent mRNA booster-dose, as well as a modest effect of boosting towards the enhancement of IFNγ$^+$ T cell responses in the lung compared to LT vaccinated individuals. Indeed, older adults not responding to vaccination have been shown to benefit from a third dose[39], and there is an obvious benefit of boosting to provide a higher degree of antibody-mediated protection from infection in the context of a high incidence of VOC[1,42]. Still, if virus neutralization is unable to completely block infection, a more robust and broader T$_{RM}$ response established in the lung of convalescent-infected individuals may have more chances of limiting disease. In this sense, polyfunctional CD107a$^+$ IFNγ$^+$ cells may contribute to infection clearance or even limit the occurrence of breakthrough SARS-CoV-2 infections vaccination, and the absence of these cells in vaccinated patients underlines the need for the development of mucosal vaccines[44], recently shown to be effective in inducing sterilizing immunity in mice[45]. The inclusion of other protein fragments, such as nucleocapsid peptides[5,8,23,27,30] in combination with mucosal routes[46] will likely contribute to the establishment of optimal memory T cells in future vaccine strategies.

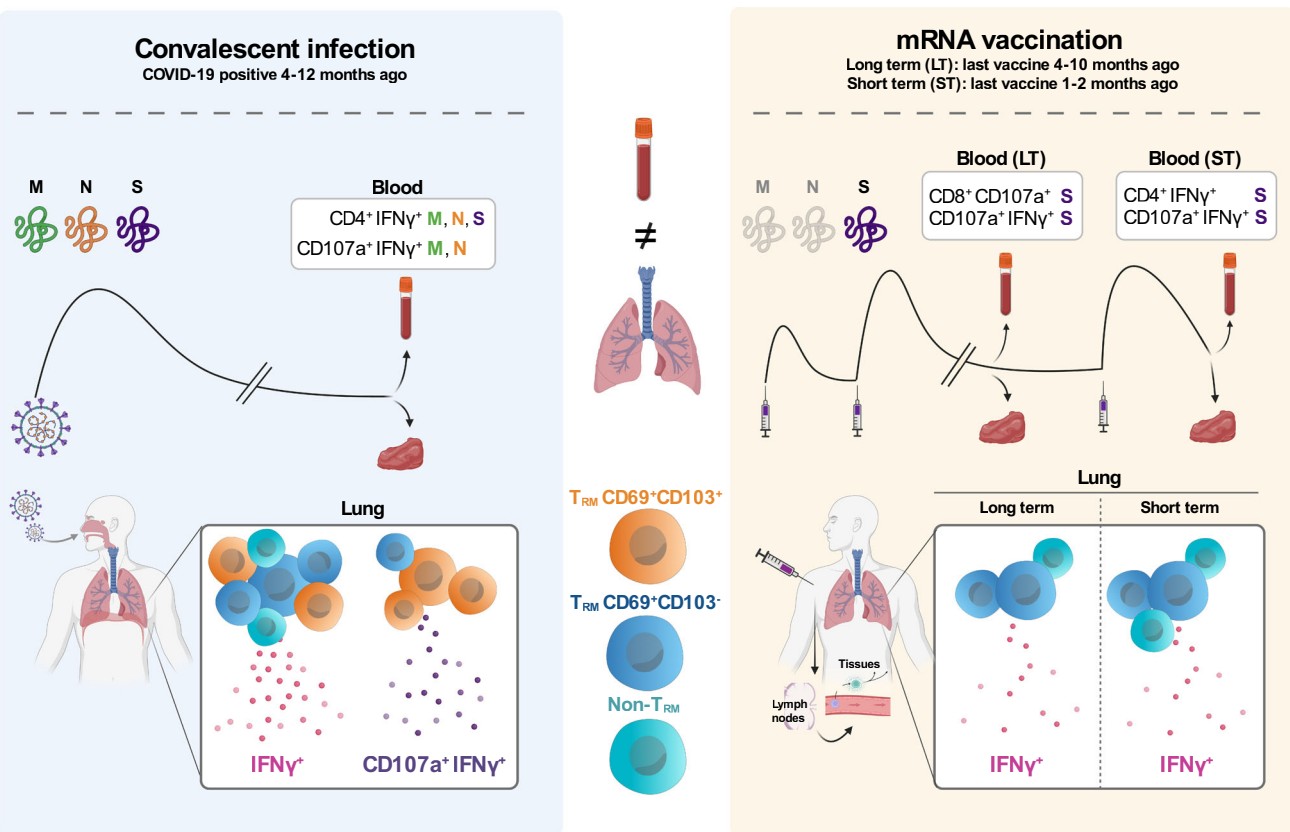

**Fig. 7 | Graphical summary of the main findings in this study.** The focus of the study was to identify T cell responses against SARS-CoV-2 in paired blood and lung tissue samples of patients that recovered from COVID-19 between 4 to 12 months ago (convalescent infection, left panel) and patients that were uninfected and mRNA vaccinated either between 4 to 10 months ago (LT long term, right panel) or between 1 and 2 months ago (ST short term, right panel). Left: Infection with SARS-CoV-2 triggers the immune system to respond against multiple viral proteins, including the M, N, and S proteins. In the blood of convalescent patients, CD4+ T cells responded to M, N, and S peptides by producing IFNγ and both CD4+ and CD8+ T cells responding to M and N peptides by producing IFNγ and CD107a. Moreover, in the lung of these patients, we found T cells with mainly T$_{RM}$

(CD69+CD103+ and CD69+CD103-) phenotypes producing IFNγ or a combination of IFNγ and CD107a. Right: mRNA-vaccinated patients are only exposed to mRNA encoding for the SARS-CoV-2 S protein. In the blood of LT vaccinated patients, CD8+ T cells responded to S peptides by producing CD107a and CD4+ and CD8+ producing IFNγ and CD107a. In the blood of ST-vaccinated patients, the response to S peptides mainly consisted of CD4+ T cells producing IFNγ and CD4+ and CD8+ producing IFNγ and CD107a. In contrast to T cell responses in the lung of convalescent-infected patients, both LT and ST mRNA-vaccinated patients showed a less prominent IFNγ response mainly produced by CD69+CD103- T$_{RM}$ cells and non-T$_{RM}$ cells. Last, CD69+CD103+ T$_{RM}$ producing IFNγ and CD107a were virtually absent in the lungs of these vaccinated patients.

## Methods

### Ethics statement
This study was performed in accordance with the Declaration of Helsinki and approved by the corresponding Institutional Review Board (PR(AG)212/2020) of the Vall d'Hebron University Hospital (HUVH), Barcelona, Spain. Written informed consent was provided by all patients recruited for this study.

### Subject recruitment and sample collection
Patients undergoing lung resection for various reasons at the HUVH were recruited through the Thoracic Surgery Service and invited to participate. Initially, a total of 49 patients, from whom paired blood samples and lung biopsies were collected, were assayed. However, based on the vaccination and/or infection status of the recruited patients, 30 (+2: HL174 and HL162) patients were finally included. Figure 1a represents a schematic summary of the study groups. Moreover, patients included in the long-term and short-term vaccination groups received multiple doses of BTN162b2 (Pfizer/BioNTech) or mRNA-1273 (Moderna) or a combination of these two vaccines. One patient included in the long-term group of this study received a combination of ChAdOx1 (AstraZeneca) and mRNA-1273 (Moderna). Supplementary Table 1 summarizes relevant information from all included patients with paired blood and lung biopsy samples.

Additionally, blood samples of convalescent infected and recently (<1 week) vaccinated (BTN162b2) healthy younger aged adults (<45 years, n = 3) were included to compare 15-mer to 9/10-mer S peptide pools and the expression of CXCR3 and CCR7 by blood T cells. For all participants, whole blood was collected with EDTA anticoagulant. Plasma was collected and stored at −80 °C (except for four patients distributed among the different groups, as indicated in Supplementary Table 1, for which this sample was not available), and PBMCs were isolated via Ficoll–Paque separation and processed immediately for stimulation assays.

### Phenotyping and intracellular cytokine staining of lung biopsies
Immediately following surgery, healthy areas from patients undergoing lung resection were collected in antibiotic-containing RPMI 1640 medium and processed as published[8]. Briefly, 8-mm³ dissected blocks were first enzymatically digested with 5 mg/ml collagenase IV (Gibco) and 100 μg/ml of DNase I (Roche) for 30 min at 37 °C and 400 rpm and, then mechanically digested with a pestle. The resulting cellular suspension was first filtered through a 70 μm pore-size cell strainer and then filtered through a 30 μm pore-size cell strainer (Labclinics). After washing with PBS, cells were stimulated in a 96-well round-bottom plate for 16 to 18 h at 37 °C with 1 μg/mL of SARS-CoV-2 peptides (PepTivator SARS-CoV-2 M, N or S (15-mer S peptide pools), Miltenyi

Biotec) in the presence of 3.3 μL/mL α-CD28/CD49d (clones L293 and L25), 0.55 μL/mL Brefeldin A, 0.385 μL/mL Monensin, and anti-CD107a (PE-Cy5, clone H4A3, #555802; 1:20) (all from BD Biosciences). Of note, for additional experiments, we used shorter 9/10-mer S peptide pools (Miltenyi Biotec) to compare the T cell response to the longer 15-mer S peptide pools that we used in all other experiments. For each patient, a negative control, cells treated with medium, and positive control, cells incubated in the presence of 0.4 nM PMA and 20 μM Ionomycin, were included. The next day, cellular suspensions were stained with Live/Dead Aqua (Invitrogen) and anti-CD103 (FITC, clone Ber-ACT8, Biolegend, #350204; 1:50), anti-CD69 (PE-CF594, clone FN50, BD Biosciences, #562617; 1:30), anti-CD40 (APC-Cy7, clone HB14, Biolegend, #313017; 1:10), anti-CD8 (APC, clone RPA-T8, BD Biosciences, #561952; 1:50), anti-CD3 (BV650, clone UCHT1, BD Biosciences, #563851; 1:166), and anti-CD45 (BV605, clone HI30, BD Biosciences, #564047; 1:50) antibodies. Cells were subsequently fixed and permeabilized using the FoxP3 Fix/Perm kit (BD Biosciences) and stained with anti-IL-4 (PE-Cy7, clone 8D4-8, eBioscience, #25-7049-82; 1:40), anti-IL-10 (PE, clone JES3-19F1, BD Biosciences, #559330; 1:10), anti-T-bet (BV421, clone 4B10, Biolegend, #644815; 1:40), and anti-IFNγ (AF700, clone B27, Invitrogen, #MHCIFG29; 1:40) antibodies. After fixation with PBS 2% PFA, cells were acquired on a BD LSR Fortessa flow cytometer (Cytomics Platform, High Technology Unit, Vall d'Hebron Institut de Recerca).

### Phenotyping and intracellular cytokine staining in blood

Freshly isolated PBMCs were labeled for CCR7 (PE-CF594, clone 150503, BD Biosciences, #562381; 1:100) and CXCR3 (BV650, clone G025H7, BD Biosciences, #353730; 1:28) for 30 min at 37 °C. After washing with PBS, PBMCs were stimulated in a 96-well round-bottom plate for 16 to 18 h at 37 °C with 1 μg/mL of SARS-CoV-2 peptides together with the same concentration of Brefeldin A, Monensin, α-CD28/CD49d and CD107a (PE-Cy5, clone H4A3, BD Biosciences, #555802; 1:20), as stated for the lung suspension above and published before[8]. For each patient, negative control and positive control were also included. After stimulation, cells were washed twice with PBS and stained with an Aqua LIVE/DEAD fixable dead cell stain kit (Invitrogen). Cell surface antibody staining included anti-CD3 (PerCP, clone SK7, #340663; 1:10), anti-CD4 (BV605, clone RPA-T4, #562658; 1:20) and anti-CD56 (FITC, clone B159, #562794; 1:50) (all from BD Biosciences). Cells were subsequently fixed and permeabilized using the Cytofix/Cytoperm kit (BD Biosciences) and stained with anti-Caspase-3 (AF647, clone C92-605, BD Biosciences, #560626; 1:33), anti-Bcl-2 (BV421, clone 100, Biolegend, #658709; 1:80), anti-IL-4 (PE-Cy7, clone 8D4-8, eBioscience, #25-7049-82; 1:40), anti-IL-10 (PE, clone JES3-19F1, BD Biosciences, #559330; 1:10), and anti-IFNγ (AF700, clone B27, Invitrogen, #MHCIFG29; 1:40) antibodies for 30 min. Cells were then fixed with PBS 2% PFA and acquired on a BD LSR Fortessa flow cytometer with BD FACSDiva software (v8.0).

### Transwell cell-migration assay

Freshly isolated or thawed PBMCs ($5 \times 10^5$ cells) and lung dissected blocks (8–9 blocks) were placed in 24-well transwell inserts (pore size 5 μm) (Sarstedt, Nümbrecht, Germany) in RPMI medium. Lower wells of the 24-well plate contained RPMI medium only or RPMI medium containing chemoattractants CCL19 (100 ng/mL), CCL21 (100 ng/mL), and S1P (10 nM) to promote cell migration, similar to a previous publication[15]. A minimum of nine transwell replicates were performed for each condition per patient sample. The plate was incubated overnight at 37 °C. The following day, lung tissue blocks in the insert were digested as described above and cells that migrated from the tissue blocks to the lower well were harvested. PBMCs in both the insert and the lower well were collected and labeled with CCR7 and CXCR3, as described above. Next, PBMC and lung single-cell suspensions were stimulated for five hours at 37 °C with combined M, N, and S SARS-CoV-

2 peptides (1 μg/mL/peptide) or 0.4 nM PMA and 20 μM Ionomycin in the presence of 3.3 μL/mL α-CD28/CD49d (clones L293 and L25), 0.55 μL/mL Brefeldin A, 0.385 μL/mL Monensin, and 5 μL/100 μL anti-CD107a-PE-Cy5. Subsequently, stimulated samples were processed for extra- and intracellular staining as described above and acquired on a BD LSR Fortessa flow cytometer.

### SARS-CoV-2 serology

The serological status of patients included in this study was determined in serum samples using two commercial chemiluminescence immunoassays (CLIA) targeting specific SARS-CoV-2 antibodies: (1) Elecsys Anti-SARS-CoV-2 (Roche Diagnostics, Mannheim, Germany) was performed on the Cobas 8800 system (Roche Diagnostics, Basel, Switzerland) for the determination of total antibodies (including IgG, IgM, and IgA) against nucleocapsid (N) SARS-CoV-2 protein; and (2) Liaison SARS-CoV-2 TrimericS IgG (DiaSorin, Stillwater, MN) was performed on the LIAISON XL Analyzer (DiaSorin, Saluggia, Italy) for the determination of IgG antibodies against the spike (S) glycoprotein.

### Pseudovirus neutralization assay

The spike of the Omicron SARS-CoV-2 was generated (GeneArt Gene Synthesis, Thermo Fisher Scientific) from the plasmid containing the D614G mutation with a deletion of 19 amino acids, which was modified to include the mutations specific for this VOC (A67V, Δ69-70, T95I, G142D/Δ143-145, Δ211/L212I, ins214EPE, G339D, S371L, S373P, S375F, K417N, N440K, G446S, S477N, T478K, E484A, Q493R, G496S, Q498R, N501Y, Y505H, T547K, D614G, H655Y, N679K, P681H, N764K, D796Y, N856K, Q954H, N969K, and L981F) (kindly provided by Drs. J. Blanco and B. Trinite). Pseudotyped viral stocks of VSV*ΔG(Luc)-S were generated following the protocol described in ref. 47. Briefly, 293T cells were transfected with 3 μg of the omicron plasmid (pcDNA3.1 omicron). The next day, cells were infected with a VSV-G-Luc virus (MOI = 1) for 2 h and washed twice with warm PBS. To neutralize contaminating VSV*ΔG(Luc)-G particles cells were incubated overnight in media containing 10% of the supernatant from the I1 hybridoma (ATCC CRL-2700), containing anti-VSV-G antibodies. The next day, viral particles were harvested and titrated in VeroE6 cells by enzyme luminescence assay (Britelite plus kit; PerkinElmer). For the neutralization assays, VeroE6 cells were seeded in 96-well white, flat-bottom plates (Thermo Scientific) at 30,000 cells/well. Plasma samples were heat-inactivated and diluted four-fold towards a concentration of 1/32 of the initial sample. Diluted plasma samples were then incubated with a pseudotyped virus (VSV*ΔG(Luc)-S$^{omicron}$) with titers of $-1 \times 10^6$–$5 \times 10^5$ RLUs/ml of luciferase activity—in a 96-well plate flat-bottom for 1 h at 37 °C, 5% $CO_2$. Next, 30,000 VeroE6 cells were added to each well and incubated at 37 °C, 5% $CO_2$ for 20–24 h. Then, viral entry was measured by the expression of luciferase. Cells were incubated with Britelite plus reagent (Britelite plus kit; PerkinElmer) and then transferred to an opaque black plate. Luminescence was immediately recorded by a luminescence plate reader (LUMIstar Omega). Viral neutralization was calculated as the reciprocal plasma dilution (ID50), resulting in a 50% reduction in relative light units. If no neutralization was observed, an arbitrary titer value of 16 (half of the limit of detection [LOD]) was reported.

### Statistical analyses and reproducibility

Flow-cytometry data were analyzed using FlowJo v10.7.1 software (TreeStar). Data and statistical analyses were performed using Prism 8.3.0 (GraphPad Software, La Jolla, CA, USA). Data shown in bar graphs were expressed as median and Interquartile range (IQR), unless stated otherwise. Correlation analyses were performed using non-parametric Spearman rank correlation. Kruskal–Wallis rank–sum test with Dunn's post hoc test was used for multiple comparisons. Friedmann test or Wilcoxon test with Dunn's post hoc test were applied for paired comparisons. A $P$ value < 0.05 was considered statistically significant.

Antigen-specific T cell data were calculated as the net frequency, where the individual percentage of expression for a given molecule in the control condition (Ctrl) was subtracted from the corresponding SARS-CoV-2-peptide stimulated conditions. No statistical method was used to predetermine sample size, as this was dependent on patient consent and eligibility to the study groups. No data were excluded from the analyses. The experiments were not randomized and the investigators were not blinded to allocation during experiments and outcome assessment, although samples were measured and analyzed in a standardized way.

### Reporting summary

Further information on research design is available in the Nature Portfolio Reporting Summary linked to this article.

## Data availability

The data supporting the findings of this study are available in the main article and its supplementary files. Source data are provided with this paper.

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

## Acknowledgements

We would like to thank all the patients who participated in the study and Drs. Julià Blanco and Benjamin Trinite for providing the plasmid encoding the omicron spike. Figures 1a, 7 and Supplementary Figs. 9a and 10a were created with BioRender.com. This work was supported by grants from Fundació La Marató TV3 (201814-10 FMTV3 and 202112-30 FMTV3, M.G.), from the Health department of the Government of Catalonia (DGRIS 1_5, to M.G. and M.J.B). M.J.B. is supported by the Miguel Servet program funded by the Spanish Health Institute Carlos III (CP17/00179). The funders had no role in study design, data collection and analysis, the decision to publish, or the preparation of the manuscript.

## Author contributions

Conceptualization, M.G.; Patient recruitment and sample collection, J.R. and V.F.; Methodology, D.K.J.P., S.G.K., A.G.R., J.R.-C., C.M., and J.E.; Investigation, D.K.J.P., S.G.K., A.G.R., and J.E.; Formal analysis, D.K.J.P., A.G.R., M.J.B., and M.G.; Writing—original draft, D.K.J.P. and M.G.; Writing—review and editing, all authors; Funding acquisition, M.G.; Supervision, M.G.

## Competing interests

The authors declare no competing interests.
