## [Peer Review File · Nature Communications]

Limited induction of polyfunctional lung-resident memory T cells against SARS-CoV-2 by mRNA vaccination compared to infectionREVIEWER COMMENTS

Reviewer #1 (Remarks to the Author):

In this work the authors investigate the presence of TRM in blood and lung after mRNA vaccination (2 or three doses) or in convalescent unvaccinated individuals. mRNA vaccination can induce strong T cell responses in peripheral blood but seems to fail to induce lung-resident TRM, as is observed in convalescent individuals. Two interesting specific patient case studies with different vaccine history are also discussed.

The outcome of these experiments is not unexpected but nevertheless it is important to investigate the extent with which peripheral T cell responses reflect the situation in the lung. Therefore the results described in this manuscript are highly relevant for the novel mRNA vaccines that have been rolled out during the COVID-19 pandemic.

The authors suggest on page 8 the following: "Together our data indicates that S-specific CD4+ T-cell responses are detectable in the lung of uninfected vaccinated patients, suggesting that mRNA vaccination against SARS-CoV-2 may potentially elicit tissue-localized protective T-cell responses already after the second mRNA vaccine dose." This claim of tissue localization of protective T cell responses is not supported by the observation that T cells are found in lung, as these T cells might reside in the blood vessels within the resected lung tissue. Unless it is clearly shown by microscopy that this is not the case. This needs to be addressed

The authors should discuss in more depth the potential effects of cancer treatment and age on the obtained results. Would the authors expect less variability if results were obtained from otherwise healthy individuals? Do the specific cancer treatments have an impact on T cell residency? Is this known from literature or other experiments?

Reviewer #2 (Remarks to the Author):

Pieren et al, evaluated antibody response and T-cell responses following SARS-CoV-2 infection and vaccination in lungs in older cancer patients. Interestingly infection demonstrated polyfunctional T resident memory cells compared with vaccine induced response in these limited number of individuals in each group. This study has several major limitations and missing important information.

MAJOR COMMENTS:

1. The title is misleading and should accurately depict the study outcome. It should mention that these are 'polyfunctional' T cells and title must say 'older cancer patients', as this study was only performed in the cancer patients.
2. Figure 1 should include a panel for study design that shows number of people in each arm, their age, gender distribution, how long after exposure the samples were collected. Also, in methods section, describe what COVID vaccine they got and what was the diagnosis and medical treatments for these cancer patients.
3. One of the major caveats of the study that there is minimal functional T cell analysis in this manuscript. Authors should perform atleast some functional T cell analysis to demonstrate that these T cells are effective against SARS-CoV-2
4. The authors mentioned that they used 15-mer peptides and not 9-10 mer peptides that are preferred for T-cell assays. They surmised that its less relevant by citing couple of previous publications, while majority of T-cell field uses 9-10 mer and have been shown to be important for performing T-cell assay. Since the T-cell assays are the critical part of the study, it will be important that T-cell assays are performed with 9-10 mer peptides to be

accurately confirm the provocative findings between infection vs vaccination shown in this study.

5. The major limitation is small number of participants in each arm and this study is not longitudinal, and the statistical analysis is almost meaningless, as none of the group has even 10 individuals. The study will be more meaningful and conclusive, if authors can atleast add some more individuals to each group so that differences/similarities in the immune response can be appropriately analyzed statistically.

6. Authors should add a figure with a schematic to describe important conclusions and their relevance of the study outcome

Reviewer #1 (Remarks to the Author):

In this work the authors investigate the presence of TRM in blood and lung after mRNA vaccination (2 or three doses) or in convalescent unvaccinated individuals. mRNA vaccination can induce strong T cell responses in peripheral blood but seems to fail to induce lung-resident TRM, as is observed in convalescent individuals. Two interesting specific patient case studies with different vaccine history are also discussed.

The outcome of these experiments is not unexpected but nevertheless it is important to investigate the extent with which peripheral T cell responses reflect the situation in the lung. Therefore the results described in this manuscript are highly relevant for the novel mRNA vaccines that have been rolled out during the COVID-19 pandemic.

The authors suggest on page 8 the following: "Together our data indicates that S-specific CD4+ T-cell responses are detectable in the lung of uninfected vaccinated patients, suggesting that mRNA vaccination against SARS-CoV-2 may potentially elicit tissue-localized protective T-cell responses already after the second mRNA vaccine dose." This claim of tissue localization of protective T cell responses is not supported by the observation that T cells are found in lung, as these T cells might reside in the blood vessels within the resected lung tissue. Unless it is clearly shown by microscopy that this is not the case. This needs to be addressed.

We thank the reviewer for the interest in our work and the useful comments to improve our manuscript. We understand the concerns raised regarding the exact tissue localization of the vaccine-induced T cells found in the lung, and the potential for blood vessel contamination. First of all, we would like to clarify that we thoroughly perfuse the lung tissue to get rid of remaining blood. However, we acknowledge that while the T_{RM} nature of SARS-CoV-2 specific T cells from the convalescent infected group is well supported by high frequencies and CD103 co-expression; low frequencies of these specific T cells in the vaccinated individuals together with low or lack of CD103 expression may still raise questions regarding their bona fide T_{RM} nature, even after considering their low T-bet expression. While microscopy analyses would be critical to demonstrate the specific location of these specific T cell in the lung parenchyma of vaccinated individuals, in situ staining of human tissues with MHC tetramers to detect SARS-CoV-2 specific T cells is, to the best of our knowledge, currently not available. Still, we performed complementary experiments to address this reviewer's concern.

To further confirm the resident nature of Ag-specific T cells detected in the lung, we exposed lung tissue blocks to chemokines CCL19, CCL21, and S1P overnight to attract non-T_{RM} out of the tissue in a transwell migration assay. CCL19 and CCL21 are chemokine-ligands attracting CCR7 expressing cells, while S1P promote egress of cells expressing S1PR1. The next day, we determined the presence of Ag-specific T cells by ICS in tissue blocks as well as in the supernatant (as shown in representative plots in **Supplementary Figure 8a**). While we were highly limited by sample availability (in particular of uninfected vaccinated individuals) and by the requirement of large pieces for these experiments, we performed these experiments in lung samples of four patients: all of them were vaccinated patients, from which two had a history of SARS-CoV-2 infection confirmed by PCR and two were infection naïve, as confirmed by antibody analyses. Considering the requirement of large samples needed for each condition, viral protein-peptide pools (M, N and S) were pooled to determine total Ag-specific T cells (**Supplementary Figure 8a**). These experiments indicate that most or all lung-derived SARS-CoV-2-specific CD4 (and CD8) T cells detected in vaccinated patients (infected or not) remain within the tissue blocks

regardless of the addition of chemokines, while that is not the case for specific cells present in blood samples (**Supplementary Figure 8a, b**).

In addition, we aimed to assess whether circulating SARS-CoV-2 specific T cells of vaccinated patients expressed molecules associated with migration towards tissue or, instead, to lymph nodes, by analyzing the expression of CCR7 and CXCR3. We previously used these markers to determine migration patterns of SARS-CoV-2 specific T cells during acute infection (**Grau-Expósito J, Sánchez-Gaona N et al. Nat Com, 2021**) and we here now analyzed the expression patterns in vaccinated individuals that had an IFN γ response above 0.01% after control subtraction (**Supplementary Figure 8c, d**). In addition, we also assessed these patterns in three recently boosted individuals (meaning 1 week after receiving the 4th mRNA-vaccine dose). Some examples of the individual plots showing the distribution of IFN γ +CD4+ T cells response based on CCR7/CXCR3 expression on total live cells are shown in **Supplementary Figure 8c**. The summary of the distribution of these responses visualized as pie charts based on different vaccinated groups showed that > 80% of these cells lack CCR7 expression. The absence of CCR7 indicates that most of circulating Ag-specific CD4+ T cells, including the ones generated after a recent 1-week 4th boost, are migrating to tissue (**Supplementary Figure 8d**). Moreover, out of these cells, a large fraction expressed CXCR3, which among other functions, could indicate a Th1 response migrating towards tissues such as the lung (**Sallusto et al, Nature, 1999 and Pejovsky et al, Front Immunol, 2019**).

These experiments as well as the corresponding methods have been now included in the manuscript (results: lines **199-220**; Methods: lines **476-491**). Furthermore, we have also intensified the discussion of this issue (lines **349-358**), referring to a recent article from the Masopust's laboratory (Künzli) in which they show that intramuscular mRNA vaccination in mice is sufficient to induce bona fide T_{RM} and also, that CD4 are characterized by CD69 expression (not CD103). Altogether, we are confident that our data now indicates that most of the specific SARS-CoV-2 T cell responses we observe in lung tissue of infected and vaccinated patients are bona fide T_{RM} cells.

The authors should discuss in more depth the potential effects of cancer treatment and age on the obtained results. Would the authors expect less variability if results were obtained from otherwise healthy individuals? Do the specific cancer treatments have an impact on T cell residency? Is this known from literature or other experiments?

Considering requests from this reviewer and reviewer 2, we have now included information regarding diagnosis and treatment in **Supplementary Table 1** and in the material and methods section (lines **424-432**).

Importantly, all patients but one were off treatment at the time of surgery. This is because surgery is often instigated by the suspicion of cancer, without a definitive diagnosis until surgery and thus, a biopsy occurs. Only one patient in the short-term vaccination group, as now stated in **Supplementary Table 1**, was under treatment with cisplatin and etoposide. Nonetheless, this patient showed high anti-S IgG (>800 AU/mL), high neutralization titers (>2048), and CD4+ T cells that responded to S-peptide stimulation in both blood and lung, at least indicating that this patient did not overly alter the group average. Thus, treatment was not a factor influencing our results. Moreover, while the biopsy in which we determined Ag-specific T cells was derived from surrounding healthy tissue, the final diagnosis (i.e. lung cancer or not) was not different among the groups (**Supplementary Table 1**).

Still, it is extremely difficult to know if less variability would occur if patients were to be healthy. In terms of infected patients, in our opinion, variability may be more influenced by infection history (disease severity, multiple infections, etc.). In terms of vaccinated only patients, it has recently been shown that neutralizing antibody responses and T-cell responses after SARS-CoV-2 vaccination in the blood of patients with solid tumors, including thoracic malignancies, do not differ from individuals without cancer (**Fendler et al. 2021 Nat Cancer; Fendler et al. 2022 Cancer Cell**). However, it has also previously been shown that patients with thoracic cancer are at higher risk of developing severe COVID-19 disease (**Grivas et al. Ann Oncol 2021, Kuderer et al. Lancet 2020**) and that, in general, T cell responses of cancer patients after vaccination against SARS-CoV-2 may be impaired (**Fendler A, et al. Nat Rev Clin Oncol, 2022**). These observations therefore may indicate that the vaccine-induced S-specific CD4⁺ T cells we found in our patients, may even contribute to protection against COVID-19 in these vulnerable patients. We have now added these references to the discussion of the manuscript and added the following on this part (lines **362-372**).

Last, in the previous version of the manuscript we showed that age negatively correlated with neutralizing capacity and antibody responses (**Supplemental Figure 1**), with degranulating-specific T cells in the lung (**Supplemental Figure 4b**), and lung S-specific CD8⁺ T_{RM} cells from Inf individuals (**Supplemental Figure 7d**). To further address the reviewers' remark, we made adjustments at several places in the manuscript, including, results (lines **98-104**) now specifying the age range of our patients (ranging from younger adults to middle aged adults to older aged adults) and further discussing the potential effect of age on the general response to vaccination, TRM establishment, and response to SARS-CoV-2 mRNA vaccines in the discussion (lines **372-380**).

Reviewer #2 (Remarks to the Author):

Pieren et al, evaluated antibody response and T-cell responses following SARS-CoV-2 infection and vaccination in lungs in older cancer patients. Interestingly infection demonstrated polyfunctional T resident memory cells compared with vaccine induced response in these limited number of individuals in each group. This study has several major limitations and missing important information.

MAJORCOMMENTS:

1. The title is misleading and should accurately depict the study outcome. It should mention that these are 'polyfunctional' T cells and title must say 'older cancer patients', as this study was only performed in the cancer patients.

We thank the reviewer for the interest in our work and the useful comments to improve our manuscript.

We have adjusted the title of the manuscript by mentioning polyfunctional. However, we are hesitant to mentioning older and cancer patients in the title for the following reasons.

In terms of "older", two of our patients were actually younger aged adults (24 and 43 years old), and the rest of the patients were either middle aged (50-65 years, n=11) or older aged (66-81 years, n=17). Therefore, if we consider the range of the vaccinated groups, now called Long-term [range 24-79] and Short-term [range 43-81], this includes younger patients and thus the title would not be concise. To clarify this further in our manuscript, we now emphasized the age range of our patients in the results section (lines **98-104**), we added these age ranges to **Supplementary Table 1** and to the new schematic summary **Figure 1a**, as requested in comment nr. 2. Moreover, we elaborated on the potential effect of age on our results in the discussion section (lines **372-380**).

Furthermore, we prefer not to add the word cancer because, not all of the samples originated from cancer patients (two were diagnosed with bronchiectasis, one with cavitory lung lesions, and one with a hamartoma). We have now specified the presence of the diagnosis lung cancer in **Supplementary Table 1**, as a percentage within each group. To add to this, most patients did not receive cancer treatment at the time of surgery. We have added additional references to the discussion on cancer patients and whether we expect T cell responses to differ as a consequence of a cancer diagnosis (lines **362-372**). While we are open for discussion, we propose the following title: Limited induction of polyfunctional lung-resident memory T cells against SARS-CoV-2 by mRNA vaccination.

2. Figure 1 should include a panel for study design that shows number of people in each arm, their age, gender distribution, how long after exposure the samples were collected. Also, in methods section, describe what COVID vaccine they got and what was the diagnosis and medical treatments for these cancer patients.

We have now included the following summary of the study design with relevant information on the groups, which is now the new Figure 1a.

Furthermore, characteristics such as gender distribution, age and exact times of sample collection after infection or vaccination were listed in **Supplementary table 1**. As requested, we have now also added the percentage of patients within each group diagnosed with lung cancer diagnosis and treatment received at the time of blood and lung sampling. Moreover, we have added the vaccine regime to the materials and methods (lines **424-432**).

3. One of the major caveats of the study that there is minimal functional T cell analysis in this manuscript. Authors should perform at least some functional T cell analysis to demonstrate that these T cells are effective against SARS-CoV-2.

Considering and reviewing major literature in the field, most studies addressing functional T cell responses perform AIM or ICS after peptide stimulation as the readout (Tarke et al, Cell, 2022; Keeton et al, Nature, 2022; Demaret et al, Front Immunol, 2021; or reviewed in Moss, Nat Immunol, 2022 and Grifoni et al, Cell Host Microbe, 2021). Further, the comparison of combining AIM+ICS protocol with the classical AIM or ICS assays, as performed by the groups of Crotty and Sette, showed no significant differences among protocols (Tarke et al, Cell, 2022). To a lesser extent, ELISPOT and tetramers are also used. While ELISPOT may be necessary for high throughput assays, tetramers which may be more informative, require HLA typing, which was

not possible for our studies which were performed in fresh samples (with also limited amount of tissue).

Furthermore, while more or less proteins can be used for assessing these responses, we included M, N and S as relevant viral proteins which we already studied for the previous paper (**Grau-Expósito J, Sánchez-Gaona N et al. Nat Com, 2021**). Additionally, aside from IFN γ and CD107a as functional markers, we also assessed the expression of IL-4 and IL-10. While we, in general, did not find differences in the production of these functional cytokines, these findings still indicate that functionally, the production of IL-4 and IL-10 are not impaired nor excessively produced. Last, also the choice of functions to address (IFN γ , CD107a, IL-4 and IL-10) were predetermined already by our previous manuscript and limited also by sample size in most cases.

Overall, considering the limited amount of sample from each biopsy, our efforts to perform the analyses on fresh samples (to avoid freezing/thawing changes in phenotype and induced cell death by the process) together with the fact that most relevant literature accepts AIM or ICS as indirect functional analyses we are confident of the interest of our methodology and results.

4. The authors mentioned that they used 15-mer peptides and not 9-10 mer peptides that are preferred for T-cell assays. They surmised that its less relevant by citing couple of previous publications, while majority of T-cell field uses 9-10 mer and have been shown to be important for performing T-cell assay. Since the T-cell assays are the critical part of the study, it will be important that T-cell assays are performed with 9-10 mer peptides to be accurately confirm the provocative findings between infection vs vaccination shown in this study.

We agree with the reviewer on the interest of performing a side-by-side comparison of 9-10 mer and 15-mer peptide stimulation to confirm that similar responses are detected when shorter peptides are used, since CD8 T cell may have been underestimated by the use of 15-mer peptides, as we discussed in the limitations. To address this point raised, we performed a side-by-side comparison between 15mer and 9-10mer S peptide pools in additional PBMC (n=8) and lung tissue (n=5) samples of patients that were either convalescent infected and vaccinated, or uninfected and vaccinated. As shown in a new **Supplementary Figure 6**, S-specific CD4+ IFN γ + responses are less frequent when shorter peptides are used. In contrast, CD8 T cells were more variable, yet only one sample (out of 8) clearly increased its frequency (**Supplementary Figure 6**). Overall, we consider that these findings do not change our message. This comparison has now been included in the results (lines **158-167**) and discussion (lines **340-348**) and **Supplementary Figure 6**.

5. The major limitation is small number of participants in each arm and this study is not longitudinal, and the statistical analysis is almost meaningless, as none of the group has even 10 individuals. The study will be more meaningful and conclusive, if authors can at least add some more individuals to each group so that differences/similarities in the immune response can be appropriately analyzed statistically.

We agree with the reviewer that our sample size is limited and may therefore limit certain statistical analyses. Recruiting new patients that fit the requirements of the groups we defined in the primary version of the manuscript has proven extremely complicated, mainly because the majority of patients we have access to have been vaccinated (now including a 4th boost) and the

majority of the Spanish population has experienced at least one SARS-CoV-2 infection (even if not recorded in their medical record).

Still, in order to address the point raised by the reviewer, we weekly screened the thoracic surgical program at our hospital to include new patients of interest. Indeed, we were initially able to include 17 additional patients, in which we performed the analyses in fresh paired lung and blood samples. Unfortunately, while most of these samples did not show SARS-CoV-2 infection on their medical records, additional antibody testing confirmed the presence of total Ig antibodies against the N protein, excluding these samples from the groups of interest. Overall, out of the 17 additional patients performed, three turned out to fit the requirements of the long-term vaccination group (2 or 3 vaccinations and SARS-CoV-2 N-antibody negative) resulting in n=10 for this group, and only one patient fit the requirements for the short-term vaccination group (3 or 4 vaccinations and SARS-CoV-2 N-antibody negative) resulting in n=6 for this group. In fact, the inclusion of these patients, instigated the change in nomenclature from Vx2 to LT (long-term, when last dose was > 6 months ago) and from Vx3 to ST (short-term, when last dose was ~1 month after sampling). The remaining 13 patients have been discarded from the main results, while some have been used for complementary analyses such as transwell experiments and epitope length comparison experiments. Addition of patients to the control group (no COVID-19, no vaccine) or the convalescent infection group (COVID-19, no vaccine) was not possible as these patients are not present anymore in our settings.

Importantly, the overall findings remain, and some have, statistically speaking, strengthened our findings and we are therefore confident that our data represents meaningful and conclusive evidence.

6. Authors should add a figure with a schematic to describe important conclusions and their relevance of the study outcome.

We have now included a summary figure (similar to a graphical abstract) as **Figure 7**. If the editorial team would prefer this figure as a graphical abstract *per se*, that would work for us too.

REVIEWERS' COMMENTS

Reviewer #1 (Remarks to the Author):

The authors have added experimental data to the revised manuscript and my concerns have been addressed.

Reviewer #2 (Remarks to the Author):

The authors have satisfactorily responded to our comments in the revised manuscript and included new data.
No additional comments.